# Antigen-Presenting Cells in the Airways: Moderating Asymptomatic Bacterial Carriage

**DOI:** 10.3390/pathogens10080945

**Published:** 2021-07-28

**Authors:** Lisa J. M. Slimmen, Hettie M. Janssens, Annemarie M. C. van Rossum, Wendy W. J. Unger

**Affiliations:** 1Laboratory of Pediatrics, Department of Pediatrics, Erasmus MC-Sophia Children’s Hospital, University Medical Centre Rotterdam, 3015 GD Rotterdam, The Netherlands; l.slimmen@erasmusmc.nl; 2Division of Respiratory Medicine and Allergology, Department of Pediatrics, Erasmus MC-Sophia Children’s Hospital, University Medical Centre Rotterdam, 3015 GD Rotterdam, The Netherlands; h.janssens@erasmusmc.nl; 3Division of Infectious Diseases and Immunology, Department of Pediatrics, Erasmus MC-Sophia Children’s Hospital, University Medical Centre Rotterdam, 3015 GD Rotterdam, The Netherlands; a.vanrossum@erasmusmc.nl

**Keywords:** respiratory tract infections, bacterial infection, asymptomatic carriage, pathogen-host interaction, antigen-presenting cells, microbiome

## Abstract

Bacterial respiratory tract infections (RTIs) are a major global health burden, and the role of antigen-presenting cells (APCs) in mounting an immune response to contain and clear invading pathogens is well-described. However, most encounters between a host and a bacterial pathogen do not result in symptomatic infection, but in asymptomatic carriage instead. The fact that a pathogen will cause infection in one individual, but not in another does not appear to be directly related to bacterial density, but rather depend on qualitative differences in the host response. Understanding the interactions between respiratory pathogens and airway APCs that result in asymptomatic carriage, will provide better insight into the factors that can skew this interaction towards infection. This review will discuss the currently available knowledge on airway APCs in the context of asymptomatic bacterial carriage along the entire respiratory tract. Furthermore, in order to interpret past and futures studies into this topic, we propose a standardized nomenclature of the different stages of carriage and infection, based on the pathogen’s position with regard to the epithelium and the amount of inflammation present.

## 1. Introduction

Antigen-presenting cells (APCs) in the respiratory mucosa are important gatekeepers that maintain homeostasis, as they prevent unwanted immunity to innocuous antigens while mounting immunity to incoming pathogens. Maintaining this careful balance is an ongoing process, considering the sheer amount of encounters between host APCs and potential pathogens. While the role and function of APCs in the context of invasive infection is extensively researched, much less is known about their behavior in instances where bacterial carriage does not lead to infection. Moreover, most literature about bacterial carriage of the respiratory tract focuses on either the pathogen side (epidemiological data and/or microbiological properties) or the adaptive (e.g., humoral) immune response. The innate immune response, which functions as a link between these two, is much less described. We hypothesize that the type and strength of APC-induced immunity will play an important role in either the clearance of bacteria, the establishing of carriage, or the progression to infection.

To investigate the role of APCs in bacterial carriage, the first challenge is to arrive at a clear definition of carriage, as several overlapping definitions are in use. We set out to formulate objective definitions for all stages on the continuous spectrum of carriage and infection, which will allow for comparison of different studies.

We will briefly outline the function of APCs in mounting immunity during invasive infection to provide context for our findings about APCs during bacterial carriage. Next, we discuss in-vitro and in-vivo studies, including human challenge models, that specifically address the behavior of airway APCs during bacterial carriage. We speculate that the nature of these host-pathogen interactions during carriage is qualitatively different from those during infection, and that bacterial carriage is therefore not merely the result of failed host defense, but rather a distinct state of host immunity. A better understanding of APC behavior during carriage will provide insight into the complex ways homeostasis is maintained in the respiratory tract, and also hint towards factors that might predispose an individual for symptomatic infection. We will discuss the next recent advances in our understanding of APC-mediated management of bacterial carriage.

## 2. Infection of the Respiratory Tract and Carriage of Respiratory Pathogens

### 2.1. The Respiratory Tract and Pneumonia

The respiratory tract is responsible for the continuous gas exchange that human life depends upon. Starting at the anterior nares, air travels through the nasal cavity and nasopharynx, down the oropharynx, through the vocal cords on to the larynx and trachea [1,2]. The larynx demarcates the upper respiratory tract (URT) from the lower respiratory tract (LRT). The trachea branches into two main bronchi for the left and right lung, which then go on to divide further still for lobes and segments, ending in smaller bronchioli, alveolar ducts and ultimately alveolar sacs and alveoli (Figure 1) [1,3]. Alveoli are small pouch-like structures lined with a single layer of epithelium, and are surrounded by a dense network of capillaries. This allows for the close proximity of blood and air that is necessary for the uptake of O_2_ and the discharge of CO_2_ [4].

Pneumonia is defined as inflammation of the lung tissue caused by one or several infectious agents and as such, it is the most severe manifestation of acute lower respiratory tract infection [5]. Recruitment of innate and adaptive immune cells and release of pro-inflammatory mediators by both the respiratory epithelium and immune cells lead to inflammation in the smallest airways, causing the alveolar sacs to fill with fluid and pus. This build-up of material hinders gas exchange, causing the host to experience symptoms such as coughing, tachypnea, dyspnea, chest pain, and hypoxia [5,6].

Despite important improvements in health care, respiratory infections still account for a major burden of disease worldwide. In low and middle income countries, pneumonia is the number one cause of death for children under five years of age [7,8], and the fourth cause of death for adults [9]. In high-income countries, respiratory infections are the prime cause of hospitalization of children as well as the most common reason for antibiotic prescription in both children and adults [10,11].

### 2.2. Pneumonia-Causing Pathogens

Pneumonia can be caused by a wide variety of pathogens, depending on age, geographical location, and underlying health issues [12,13]. Large studies in both adults and children with respiratory infections indicated that in at least 70% of the individuals a virus was the cause [14,15]. In the majority of cases human rhinovirus, human metapneumovirus, and Influenza A or B was detected. Respiratory syncytial virus was predominantly found in children under 5 years of age [15]. Bacteria were found in about 20–30% of children, with *Mycoplasma pneumoniae* and *Streptococcus pneumoniae* currently being the main bacteria causing pneumonia [15]. In adulthood, *S. pneumoniae* is widely reported as the most common bacterial pathogen. Other bacterial pathogens found in both children and adults include *Haemophilus influenzae, Moraxella catarrhalis, Staphylococcus aureus,* and intracellular pathogens such as *Legionella pneumophila* and Chlamidophila spp [16,17,18]. 

In addition to the aforementioned bacteria found in immunocompetent hosts, more rare bacteria are found in individuals with underlying lung disease, as well as fungi and parasites [19,20]. For instance, *Pseudomonas aeruginosa* is frequently found in cystic fibrosis (CF) patients while being a relatively rare occurrence in healthy subjects [21].

It should be noted, however, that most bacteria that have the capacity to cause pneumonia, only rarely do so. This is made apparent by the fact that the asymptomatic carriage rate of respiratory pathogens in the URT is many-fold higher than the rate of severe infection. When using sensitive PCR-based methods, at least one potential pathogen can be detected in 34–60% of healthy adults [22,23,24,25,26], with even higher carriage rates found in infants and children [27,28,29]. The estimates for yearly incidence of community-acquired pneumonia (CAP) range from 1.1 to 29.0 per 10,000 individuals, depending on the region [30]. This contrast in numbers indicates that most host-pathogen encounters do not lead to infection.

Acknowledging the fact that the majority of respiratory infections are caused by viruses, from here onward this review will focus specifically on bacterial carriage and infection. Although the host immune response to viral infections overlaps in many ways with defense mechanisms against bacteria, the pathways are quite distinct, and management of asymptomatic viral carriage is outside of the scope of this article.

### 2.3. Defining Carriage and Infection

Many respiratory micro-organisms can be detected in the respiratory tract of healthy, asymptomatic individuals [22,23,24,25,26,27,28,29]. Traditionally, the presence of a microbe on a mucosal surface that does not cause disease in the host, is defined as “carriage”, “asymptomatic carriage”, or “colonization”. In literature however, this sometimes refers to any bacterial presence found during cross-sectional sampling (mostly in the URT), and sometimes to prolonged presence after bacterial acquisition. While it has been shown that bacterial carriage can persists for months [31,32,33,34,35], it is unknown whether there is a minimum amount of time that bacteria should be present at the epithelium for a host to be regarded as “carrier”. Complicating matters even further, terms such as “subclinical infection”, “pre-clinical infection”, and “asymptomatic infection” are also used interchangeably to describe similar phenomena, sometimes even referring to mild inflammation or a “low grade infection”. This variety of definitions complicates the interpretation and comparison of studies investigating asymptomatic carriage. Therefore, before discussing available literature on APC behavior during carriage, we will first propose an objective set of definitions to serve as a framework to compare the findings of this literature. As our definition is based on the pathogen’s position with regard to the epithelium, and therefore is applicable along the entire respiratory tract, we purposefully chose not to distinguish between URT and LRT carriage in our discussion.

The asymptomatic presence of a potential pathogen in the RT can be divided into distinct phases. Firstly, it starts with the acquisition of bacteria: the introduction of a micro-organism at the airway mucosa of a host. Secondly, once bacteria have passed the mucus layer and adhere to the epithelial surface, they can establish a stable presence on the mucosa to obtain nutrients needed to survive (Figure 2). Bacterial growth during carriage amplifies bacterial density, potentially increasing the likelihood of transmission to a new host. 

Besides being the main source for horizontal spread in the community [36,37], asymptomatic carriage of bacteria is also considered to be the first step in the pathogenesis of infection. Infection is defined as the process in which a pathogen enters the tissue underlying the respiratory epithelium and multiplies. This implies that a pathogen has to breach the epithelial barrier and cross the basement membrane for infection to occur. This is in contrast to events during carriage, where the bacteria are situated on the epithelial surface without disrupting the epithelial barrier. Based on the outcome of the interaction with the host, different stages of infection may be discerned (see Figure 2). We propose that in each of these infection stages, the degree of pathogen multiplication, elicited immune response and damage to the host tissue is different, and together result in a certain degree of inflammation.

Clinical signs and symptoms arise from inflammation in the airways. Inflammation is instigated upon bacterial sensing by the airway innate immune system, i.e., the respiratory epithelium and phagocytes, via a diverse array of pattern recognition receptors such as Toll-like receptors (TLRs) [38]. Consequently, these cells produce chemokines and cytokines, which recruit and activate immune cells such as monocytes, neutrophils, and lymphocytes, as well as act on nearby endothelium of blood vessels. These blood vessels become leaky and permit exit of immune cells to the affected tissue [39], as well as extravasation of fluids. When inflammation is low or such that it leads to unapparent or mild symptoms, infection is referred to as “subclinical infection”. In case inflammation is more severe, it will result in moderate or severe symptoms and is referred to as “clinical infection”. As mentioned before, the term “subclinical infection” is often used to refer to states of “carriage” or “colonized individuals”. However, as carriage or colonization is defined by the absence of epithelial breach, a “subclinical infection” would thus refer to a stage of (mild) infection, hence the term “subclinical”. The elicited host response is insufficient to cause noticeable symptoms, with the net result being subclinical disease. As explained above, the absence of noticeable clinical symptoms does not imply that inflammation is absent. The interaction of bacteria with the host epithelium and/or DCs via their protrusions may induce cytokine production, however at this level it is not enough to evoke clear symptoms.

## 3. Progression from Carriage to Infection

Carriage is assumed a requisite precursor of invasive disease by many respiratory tract pathogens. This is substantiated by prospective studies on *S. aureus* infections demonstrating that a higher frequency of infected patients were previous carriers compared to non-carriers [27,40,41,42,43]. Further evidence comes from studies showing that decolonizing of nasal carriers of *S. aureus* on hospital admission reduced the number of surgical-site *S. aureus* infections [44]. Together, these studies emphasize carriage as a necessary predisposing condition for infection. Importantly, findings from several large prospective studies on the relationship between carriage and infection in the same study subject suggest that the main predisposing event for disease is the acquisition event rather than the prolonged carrier state. For example, non-carriers who acquire bacteremia by an exogenous *S. aureus* were shown to have a four-fold increased mortality rate compared with *S. aureus* nasal carriers [44,45]. In addition, children with pneumococcal acute otitis media were found to have a significantly lower frequency of pneumococcal carriage specific to the serotype/group causing the disease in the weeks preceding respiratory infection, compared with children who also carried pneumococci during a respiratory infection, but did not have pneumococcal acute otitis media. These studies thus confirm the role of recent acquisition of the disease-causing serotype [40]. Moreover, they underline that infection is not simply “carriage that has gotten out of control” but rather emphasize that the acquisition of a new pathogen can be a decisive moment that can determine whether carrier status or infection ensues.

It has been speculated that the development of disease is related to bacterial load. Adult pneumonia patients were shown to have more dense pneumococcal colonization than asymptomatic carriage controls, suggesting that the transition from asymptomatic carriage to disease may happen at a critical bacterial density [46]. However, in other studies *S. pneumoniae*, *H. influenzae,* and *M. pneumoniae* densities in the URT carriers were found to be highly variable over time, and *M. pneumoniae* for instance was found at similar densities in both pneumonia patients and asymptomatic carriers [29,47,48]. The causal relationship between bacterial density and symptomatic infection remains unclear and might be different depending on the pathogen. Nonetheless, these findings suggest that progression from carriage to infection is not merely a matter of bacterial overgrowth but rather the result of host-pathogen interplay. This not only hints at an important role for innate immune cells, and APCs in particular, in managing bacterial presence on the respiratory epithelium, but also highlights that the quality of this interaction will lead to progressive infection. Understanding the interactions between respiratory pathogens and airway APCs that result in asymptomatic carriage, will provide better insight into the factors that can skew this interaction towards infection, and how to possibly interfere with this process.

It is likely that the same factors that influence whether bacteria can persist and even replicate on the respiratory epithelium, will also influence the progression to invasive disease. Besides pathogen-specific factors, these include the host mucosal immune system and other bacteria, including the microbiome.

## 4. Airway APCs in Infection and Carriage

### 4.1. APC-Pathogen Interactions during Infection

Any pathogen entering the airway has to overcome several host defense mechanisms in order to establish a stable presence [5,49]. Many of these pathogens are physically removed through mucocilliary clearance, combined with coughing or sneezing. The mucus layer contains a variety of compounds with antimicrobial activity, such as antimicrobial peptides and defensins, and pathogen-specific immunoglobulins (Ig) such as secretory IgA [36].

The epithelium of the respiratory tract separates the airway lumen from lung tissue and plays an essential role in the defense against micro-organisms. Adhesion of pathogens to the respiratory epithelium can elicit the production of cytokines and chemokines, which recruit and activate neutrophils and monocytes [50]. During homeostasis, however, the most prevalent immune cells in the lamina propria, the tissue underlying the respiratory epithelium, are tissue-resident airway macrophages and dendritic cells (DCs), and patrolling monocytes [51,52,53]. Macrophages can be divided into alveolar macrophages (AMs) and the tissue-resident interstitial macrophages, of which AMs are the most well researched due to their accessibility for sampling through bronchoalveolar lavage. AMs reside in the alveoli, where their frequency is estimated around 1 AM per 3 alveoli [54]. Compared to DCs, AMs are considered to have a relatively high phagocytic capacity, but a lower antigen-presenting activity [55]. Apart from possible movement between alveoli through Kohn’s pores, they are considered non-migratory. In addition to their ability to mount a pro-inflammatory response, AMs play a crucial role in the clearance of pathogens and dead cells, and resolution of inflammation [55]. Depending on the pathogens or stimuli encountered, AMs can differentiate towards pro-inflammatory cytokine-producing M1 or more anti-inflammatory cytokine producing and tissue repair-promoting M2 phenotypes. These phenotypes are not fixed, and macrophages can shift between these “extremes” depending on environmental stimuli [56]. 

In contrast to AMs, DCs are not present in the alveoli but in the interstitial tissue. DCs can be found along the branching airways, yet the density of lung DCs decreases upon descending the airways, with DCs being more frequent in the peripheral bronchial region than in the airways close to the alveoli [57,58,59]. While DCs are typically localized beneath the respiratory epithelium, with their dendrites they can protrude through the epithelium and sample the airway lumen for antigens. DCs can internalize either whole pathogen or bacterial particles, which triggers their maturation, i.e., upregulation of costimulatory and chemokine receptors. During this process, DCs migrate to a draining lymph node where they can interact with pathogen-specific T-cells and B-cells and induce a strong effector response [60,61,62]. DCs are therefore considered the link between innate and adaptive immunity. DC subsets in the airways are similar to those in peripheral blood: CD1c^+^ and CD141^+^ myeloid DCs (also called conventional DCs, cDC1s and cDC2s respectively) and plasmacytoid DCs (pDCs) [52,59,63]. While all DC subsets have the capacity to present antigen to T cells, they perform different immune functions. cDCs most efficiently promote naive T cell activation. Priming of CD8^+^ T cells in order to combat intracellular bacteria and viruses is mostly dependent on the cDC1 subset that has superior cross-presenting capacities, while cDC2 cells are more prone at inducing Th17 and Th2 responses [64,65]. In comparison, the antigen presentation potency of pDCs is poor. Instead, pDCs are hallmarked by the rapid production of IFN-I during viral infections [66,67].

A third phagocytic cell population that can be detected in the airways are the monocytes. In steady state, the lungs are constantly patrolled by monocytes [68]. As mentioned above, during an infection, these cells are recruited in high numbers from the peripheral blood into the affected tissue and during acute inflammation also into the alveolar space. Recruited and activated monocytes are referred to as inflammatory monocytes, which have a high capacity for producing pro-inflammatory cytokines and chemokines, further promoting local inflammation [69]. Additionally, in response to local cues in the inflamed tissues, monocytes can differentiate into either monocyte-derived macrophages or DCs, which display properties distinct from their tissue-resident counterparts [51,52,68,70,71]. 

### 4.2. APC-Pathogen Interaction during Asymptomatic Carriage

The majority of the time, host immunity in the respiratory tract tends towards a tolerogenic environment, to prevent continuous inflammation that would interfere with the respiratory tract’s main function of gas exchange. The behavior of respiratory APCs in the context of invasive infection is well established. In contrast, much less is known about the role of APCs and their behavior in the context of asymptomatic carriage. Comparing studies, both with animals and human subjects, is difficult due to the variability in definitions of the states of carriage/colonization, subclinical and clinical infections. Human challenge models provide valuable insight into the host response to asymptomatic carriage, most often using the common respiratory pathogens *S. pneumoniae* and *S. aureus*. 

Using such a human challenge model, nasal carriage of *S. pneumoniae* was shown to induce an increase of IL17A^+^ T cells in bronchoalveolar lavage as compared to non-colonized individuals [72]. Th17-derived IL-17 stimulated alveolar macrophages (AMs) to increase the uptake and killing of pneumococci, suggesting that URT carriage of *S. pneumoniae* has a significant enhancing effect on lung immunity. This aspect was further investigated in a study showing that the enhanced phagocytic capacity of AMs from *S. pneumoniae* carriers lasted for 2–3 months before returning to the same level as AMs from carriage-negative controls. Notably, this enhanced uptake was also seen for other pathogens such as *S. aureus* and *S. pyogenes*, suggesting a broader state of poise [73].

In a human challenge model for *Staphylococcus aureus*, rapid clearance (defined as a negative culture within 9 days) correlated with an increase in Th1-type cytokines in nasal lavage, but interestingly not IL-17 [74]. This might suggest that a Th17 response is more associated with persistent carriage instead of rapid clearance. Additionally, nasal *S. aureus* persistence correlated with high baseline levels of MIP-1β, IL-6, and IL-1β, combined with an absence of upregulation of pro-inflammatory cytokines upon inoculation. These findings were confirmed in a similar macaque model for asymptomatic *S. aureus* carriage [75]. The effect of pre-existing differences in host immune environment at the time of pathogen acquisition fits with the clinical observation that pre-existing inflammation can predispose the host for carriage or infection with a secondary pathogen.

It should be noted that in the aforementioned human challenge studies, the control group comprised subjects that did not establish *S. pneumoniae* carriage after a nasal challenge. This common set-up for human challenge studies poses some interesting questions with regard to their interpretation, as we are not so much comparing a group that has *S. pneumoniae* carriage versus a group that does not, but rather a group that rapidly cleared *S. pneumoniae* versus a group that did not. It might be speculated that the increased phagocytic activity of AMs in the persistent carriers could be a compensatory response when initial bacterial clearance does not occur. Additionally, apoptosis of AMs after bacterial ingestion and killing is required for adequate bacterial density control by AMs [76,77]. In a murine model for a low-dose *S. pneumoniae* challenge, in which mice cleared the bacteria within 96 h with no outward signs of illness and only minor histopathological changes, an increase of apoptotic AMs in BAL fluid was observed [76]. Inhibition of apoptosis-mediated killing through a caspase inhibitor resulted in higher bacterial density in the lung and increased neutrophil recruitment.

The early kinetics after pathogen acquisition, that determine whether or not carriage is established, were investigated by collecting repeated samples in the first hours after nasal pneumococcal challenge [78]. Rapid clearance was associated with a higher number of neutrophils (but notably no other myeloid cells) in the nasal epithelium at baseline, and increased cytokine production compared to subjects who became culture positive. While this study did not focus on the function of APCs, the rapid cytokine induction suggests a role for tissue-resident (as opposed to recruited) immune cells in the very early stage of acquisition and colonization, working in concert with the respiratory epithelium to mount an almost instantaneous response.

Gaining an understanding of APC behavior in bacterial carriage remains difficult and often needs to be extrapolated from downstream effects such as cytokine and chemokine production, because direct sampling of respiratory tissue is rare. In one such study, nasal biopsies were collected after a nasal *S. pneumoniae* challenge [79]. No differences were found in the abundance of myeloid cells, but functional results regarding macrophages or DCs were not further specified.

Although outside of the scope of this review, we acknowledge that there are many factors involved in the function of macrophages and DCs. Genetic predisposition, age, host comorbidity and infectious history, and environmental factors have all been demonstrated to affect APC function [80,81,82,83]. 

### 4.3. APC Function and the Respiratory Microbiota

The term “microbiota” refers to the assembly of living micro-organisms (bacteria, archaea, fungi, protozoa, and algae), that occupy a specific ecological niche. In the context of a human host, several distinct niches are widely accepted, such as the respiratory tract, gastro-intestinal tract, and skin. Most of these micro-organisms are routinely found on the epithelial surfaces of any healthy individual without causing pathology. In fact, many micro-organisms form a symbiotic relationship with the host, for instance in the gut where microbiota are essential for proper nutrient digestion. The term “commensal bacteria” or “commensals” for short is often used interchangeably when referring to bacterial microbiota. The term microbiome encompasses both the living microbiota as well as all organic and inorganic compounds, that combine to form an ecological niche [84]. The establishing of an individual’s URT microbiome commences straight after birth [85,86], and continues to undergo changes during the course of one’s life, influenced by environmental exposure and host condition. Although the URT is a continuous system, distinct microbiotic niches have been identified, ranging from common skin bacteria near the nostrils, to more anaerobic species in the nasopharynx (considered the preferred niche for many pathogens) and paranasal sinuses and oral bacteria in the oropharynx [87]. While the lungs were long thought to be a sterile environment, it has become clear that a wide variety of microorganisms can be detected in the lower airways [87,88,89,90]. The microbiota found in the lung resemble those in the URT and those of the oropharynx in particular, and are believed to be caused by continuous micro-aspirations. While micro-aspiration is common in healthy subjects [91,92], it is more frequent in those with (chronic) lung disease, including pneumonia, chronic obstructive pulmonary disease (COPD) and asthma [93,94,95]. Together, these data indicate that both in health and disease, the LRT epithelium and innate immune cells are continually exposed to bacteria. Understanding the interactions between APCs and microbiota during homeostasis will provide insight into how APCs respond to pathogens during asymptomatic carriage.

The association between changes in the respiratory microbiome and development of respiratory disease has been well-established, and appears to be a two-way interaction. A diverse microbiome is associated with lower infection rates [96,97], and a lower biodiversity is seen in individuals that are more prone to RTIs [98,99,100,101]. Vice versa, an episode of inflammation or even the asymptomatic acquisition of a new pathogen can impact the composition of the microbiota [102,103,104,105]. Even though it could be argued that every individual’s microbiome is to some extent unique, broad patterns do emerge, and microbiomes are often compared by categorizing them into different profiles, based on the most prominent organisms found. 

### 4.4. Commensal Bacteria Prime Airway APCs

Data from both human and mice studies indicate that the presence in the lung of URT-derived bacteria such as Prevotella spp and/or their metabolic products regulates a tonic level of airway inflammation and prolonged Th17 immune activation [106]. IL-17 has been show to play a central role in the clearance of pathogenic airway bacteria [107]. More recent studies indeed showed that the commensal-induced Th17 response protected mice from *S. pneumoniae* associated infection and mortality [108].

Together, these data may implicate an immunoprotective role of episodic micro-aspiration of oral microbes in the regulation of the lung immune phenotype and mitigation of host susceptibility to infection with lower-airway pathogens, by priming the lung innate immune cells towards a Th17 response. Alternatively, one could speculate that the upregulation of pro-inflammatory cytokines through repeated episodes of oral microbe aspiration leads to chronic inflammation and could predispose the lungs to more severe inflammation, ultimately resulting in structural damage and reduced lung function.

An in-vitro study on functional effects of exposure of airway DCs to commensal bacteria, such as Prevotella spp., revealed that these types of bacteria induced lower levels of the T cell-polarizing cytokines IL-23, IL-12p70, and IL-10 in monocyte-derived DCs than pathogenic bacteria such as *H. influenza* [109]. In general, pathogenic proteobacteria induced 3–5-fold higher amounts of these cytokines than the commensal bacteria. It can be speculated that a change in the IL-23/IL-12 balance will translate into the induction of a predominant Th17-type response by the DCs as IL-23 is the key mediator of Th17 cell differentiation [108,110]. Further, coculture experiments with both commensals and pathogens revealed that Prevotella spp. reduced *H. influenza*-induced IL-12p70 but not IL-23 production in DCs by approximately 50%. This indicates that commensal bacteria of the airways, similarly to the gut, may exhibit properties that enable modulation of the APC response to specific pathogenic bacteria.

Importantly, a higher abundance of oral bacteria in the lung also affects the function of AMs, resulting in a blunted pro-inflammatory response [106]. AMs from bronchoalveolar lavage (BAL) samples with a relatively high abundance of oral bacteria displayed a decrease in macrophage-derived chemokine (MDC), IL-6 and GM-CSF production when stimulated with LPS. It could be speculated that this blunting effect of the microbiota counters a more potent TLR4 signaling by pathogens, and hereby protects against undesirable inflammation. 

Contrastingly, the protective effect of *S. aureus* carriage against influenza-mediated lung injury was not dependent on TLR4 but rather on TLR2 signaling. *S. aureus* priming resulted in an upregulation of TLR2 in AMs, and both AMs and recruited monocytes differentiated towards a pro-resolving M2 phenotype [111]. A comparable type of modulation of AM function is seen after severe pulmonary infection and inflammation. While asymptomatic carriage of bacteria enhances the phagocytic capacity of AMs, severe infection and inflammation was followed by a sustained period with reduced phagocytic activity (defined as “immune paralysis”), lasting several months [112]. Instead of being caused by immune cell exhaustion, this paralyzed state appears to be actively promoted by the lung environment after an episode of inflammation, and might serve as a protection against protracted inflammation and resulting lung damage. 

A contributing factor to the differences seen in TLR stimulation between commensals and pathogens is the nature of their TLR interaction and subsequent induction of an inflammatory response. LPS can be either penta- or hexa-acetylated, with hexa-acetylated LPS being a 100-fold more potent immune-stimulant when bound to TLR than the penta-acetylated variant [113]. The more potent hexa-acetylated LPS is found mostly on Gammaproteobacteria, a phylum that contains many potentially pathogenic bacteria, while commensal bacteria typically only express the penta-acetylated variant [114]. The fact that many pathogens express LPS that more efficiently induces inflammation, and thereby possibly expedite their own eradication, appears counterintuitive at first, but might actually serve a purpose to these pathogens. Many of the antimicrobial compounds produced by inflammatory immune cells, such as reactive oxygen species (ROS) and reactive nitrogen species (RNS) can be sourced for anaerobic respiration, provided that the bacteria are equipped with the right machinery. As it so happens, most Gammaproteobacteria, including *P. aeruginosa*, encode these pathways. This suggests a perpetuating loop, where bacteria expressing the most immune-stimulatory molecular patterns, also benefit metabolically from an inflamed environment. Pathogens run the risk of provoking an inflammatory response that might lead to their own clearance from the host, but benefit from a level of low-grade inflammation. It could be speculated that this is a contributing factor in the co-occurrence of chronic bacterial carriage and chronic inflammation that is often observed in chronic lung disease such as COPD and CF. In many of these diseases, distinct changes in microbiota are observed, with a higher prevalence of potential pathogens such as *H. influenza* and *P. aeruginosa*, which may in part be attributed to widespread use of antibiotics in this group of chronic lung disease patients [115,116], as antibiotic use has been shown to affect microbiome composition both in short and long term [117,118]. This possible incentive for pathogens to create an inflammatory environment further highlights the role of commensal bacteria in modulating host immune response, for example the blunting of TLR4 response associated with oral bacteria [106]. 

## 5. Concluding Remarks and Future Perspectives

Asymptomatic carriage of potentially pathogenic bacteria is common, and most of the time does not result in infection. Here we provide an overview of the available literature on the role of respiratory APCs during asymptomatic bacterial carriage and how this is influenced by the local microbiome. Many overlapping definitions of the term carriage are being used. In order to interpret and compare studies involving bacterial carriage, we have proposed a nomenclature where we integrated the position of the pathogen and the absence or presence of symptoms. We concluded that there are inherent differences in the interactions between APCs and either pathogens or microbiota. 

Looking at the available evidence from controlled challenge studies modeling asymptomatic carriage, it appears that the first hours and days after initial acquisition are the most decisive in determining whether a new pathogen will establish carriage. Immediate protective effects from previous exposure by both commensals and pathogens is closely linked to AM phagocytic, bactericidal, and apoptotic capacity. In addition to rapid clearance, DCs are needed for induction of a protective adaptive response to manage both current and future carriage. Early eradication seems to hinge on a rapid induction of a Th1 response. When this initial response does not clear the new pathogen, a more long-term Th17-type response becomes more prominent in managing bacterial presence. Carriage with both pathogenic and commensal bacteria can prime airway APCs in complex ways, enhancing the phagocytic capacity of AMs while at the same time dampening immune responses in a way that appears to provide protection against severe inflammation. Interestingly, commensal bacteria also elicit a low level of Th17-type inflammation, and it remains to be seen whether this low-grade inflammation is beneficial to the host.

Studies into the interaction between asymptomatically carried bacteria and respiratory APCs would be valuable. More specifically, the host-pathogen interactions in the first hours after acquisition remain largely unexplored, but appear to play a key role. Are there specific APC phenotypes that predispose the host to either fast clearance, persistent carriage, or symptomatic infection? What skews these APCs towards these phenotypes, and what points in this process can be used to intervene in the development of infection? Deeper insight into the process of carriage from the perspective of host immunity will inevitably lead to better understanding of the pathogenesis of infection, providing new avenues for diagnostic tools and treatments. 

## Figures and Tables

**Figure 1 pathogens-10-00945-f001:**
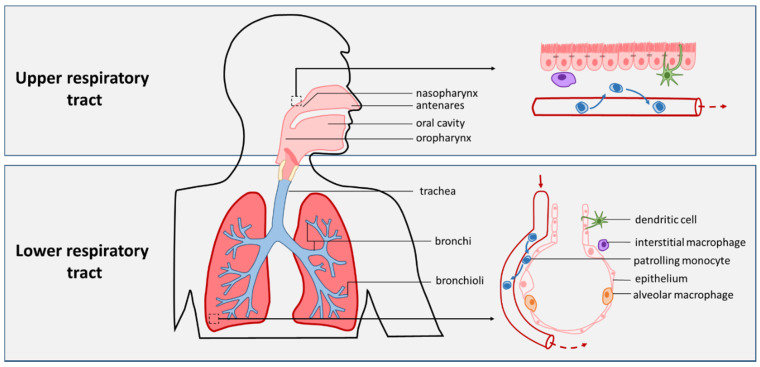
Anatomy of respiratory tract and position of antigen-presenting cells. The upper respiratory tract is comprised of the antenares, nasal cavity, nasopharynx, oral cavity, and oropharynx. The larynx demarcated the transition from upper respiratory tract to lower respiratory tract. The trachea branches into bronchi, bronchiole, and eventually terminal bronchiole, which lead to alveolar sacs. Each alveolar sac contains several alveoli, pouch-like structures lined with a single layer of epithelium. Alveoli are covered by a dense network of capillaries to allow for gas exchange. Dendritic cells can be found along the entire respiratory tract up until the terminal bronchioli. Alveolar macrophages reside in the alveolar lumen, while interstitial macrophages reside in the interstitium. Monocytes from the blood can leave and re-enter the bloodstream.

**Figure 2 pathogens-10-00945-f002:**
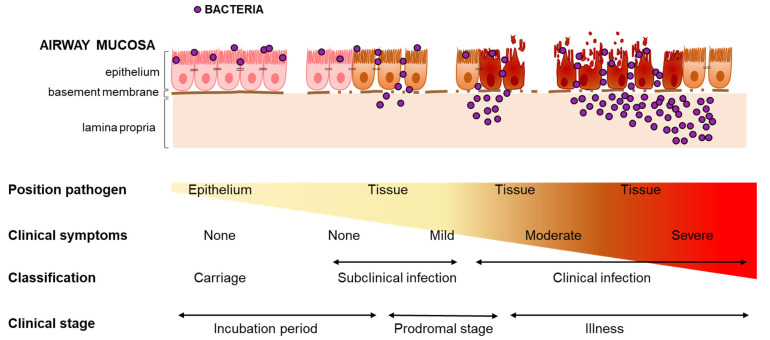
Proposed classification of carriage of and infection with bacteria in the respiratory tract in relation to clinical symptoms. Bacteria entering the respiratory tract and that have evaded the first line of defense can establish themselves at the epithelium of a healthy host. Balance with the host allows continued presence over time leading to carriage. Events that lead to breaching the airways and invasion of the underlying tissue are referred to as infection. Depending on the degree of invasion, i.e., whether the bacteria can multiply strongly, infection will induce different degrees of inflammation in the host. The degree of inflammation will direct the severity of symptoms a host experiences, and herewith regulates whether an infection is “subclinical” or “clinical”. The different stages of carriage and infection overlap with some of the clinically defined phases of an infection: incubation period, prodromal stage, and illness. The incubation period covers the time from initial contact with the pathogen to the appearance of the first symptoms. The prodromal stage is defined by vague feelings of discomfort and nonspecific complaints, while illness is hallmarked by specific signs and symptoms such as fever, dyspnea, and coughing.

## Data Availability

Not applicable.

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
