# Peer review of "Antigen-Presenting Cells in the Airways: Moderating Asymptomatic Bacterial Carriage"

_pathogens, 2021, doi:10.3390/pathogens10080945_

Round 1
Reviewer 1 Report
The manuscript entitled “Antigen-presenting cells in the airways moderating asymptomatic bacterial carriage” by Slimmen et al addresses the question of the parameters that can lead to carriage versus infection in the respiratory tract. The paper provides a comprehensive background required for understanding the anatomic basis of the respiratory tract as well as antigen-presenting cells of the respiratory system. It also provides the definitions required for understanding carriage versus infection. The data are supported with clinically-relevant examples of studies in the field. There are few minor issues that should be addressed: 1. In section 3.1, it should be noted that DC can also engulf both whole pathogens and bacterial particles to be processed and presented in the LN. 2. The description of DC subpopulation is limited (line 274-275) and should be expanded. It should be emphasized that these subpopulations are different in function, can prime different T cell subsets and some (pDC) are less specialized in antigen presentation and are a main source of IFN I. Cross-presentation is essential for CD8 T cell priming, the process is mainly attributed to cDC1. 3. Please add a description of inflammatory monocytes in the context of host response to infection. 4. Please address the outcome of repeated antibiotic treatment on the respiratory microbiome and recurring infections 5. Line 425: the word "found" is written twiceAuthor Response
Please see the attachment.

Reviewer 2 Report
The intention of the authors was to present a very interesting topic of the participation of antigen presenting cells in cases of asymptomatic bacterial carriage. However, the article has many imperfections and requires thorough improvement. Unclear and confusing for the reader is which part of the respiratory tract is covered by the review. The presented purpose of the work is discussed by the authors only briefly, so the work does not fully explain the established assumption. The first fundamental question is, what does it bring to the literature on the subject compared to other works on this topic? Second, what was the criterion for selecting literature to achieve this goal? What methodology was used? It should be clearly presented in the manuscript. I am asking to verify the microbiome's definition following the latest international committees' arrangements and provide the references for the given definition.
Round 2
Reviewer 2 Report
All queries have been addressed.